# Inverse Filtering for Hidden Markov Models

**Robert Mattila**
Department of Automatic Control
KTH Royal Institute of Technology
`rmattila@kth.se`

**Cristian R. Rojas**
Department of Automatic Control
KTH Royal Institute of Technology
`crro@kth.se`

**Vikram Krishnamurthy**
Cornell Tech
Cornell University
`vikramk@cornell.edu`

**Bo Wahlberg**
Department of Automatic Control
KTH Royal Institute of Technology
`bo@kth.se`

## Abstract

This paper considers a number of related *inverse filtering* problems for *hidden Markov models* (HMMs). In particular, given a sequence of state posteriors and the system dynamics; *i)* estimate the corresponding sequence of observations, *ii)* estimate the observation likelihoods, and *iii)* jointly estimate the observation likelihoods and the observation sequence. We show how to avoid a computationally expensive *mixed integer linear program* (MILP) by exploiting the algebraic structure of the HMM filter using simple linear algebra operations, and provide conditions for when the quantities can be uniquely reconstructed. We also propose a solution to the more general case where the posteriors are noisily observed. Finally, the proposed inverse filtering algorithms are evaluated on real-world polysomnographic data used for automatic sleep segmentation.

## 1 Introduction

The *hidden Markov model* (HMM) is a cornerstone of statistical modeling [1–4]. In it, a latent (i.e., hidden) state evolves according to Markovian dynamics. The state of the system is only indirectly observed via a sensor that provides noisy observations. The observations are sampled independently, conditioned on the state of the system, according to observation likelihood probabilities. Of paramount importance in many applications of HMMs is the classical *stochastic filtering problem*, namely:

> *Given observations from an HMM with known dynamics and observation likelihood probabilities, compute the posterior distribution of the latent state.*

Throughout the paper, we restrict our attention to discrete-time finite observation-alphabet HMMs. For such HMMs, the solution to the filtering problem is a recursive algorithm known as the *HMM filter* [1, 4].

In this paper, we consider the inverse of the above problem. In particular, our aim is to provide solutions to the following *inverse filtering problems*:

> *Given a sequence of posteriors (or, more generally, noisily observed posteriors) from an HMM with known dynamics, compute (estimate) the observation likelihood probabilities and/or the observations that generated the posteriors.*

To motivate these problems, we give several possible applications of our results below.

**Applications**   The underlying idea of inverse filtering problems ("*inform me about your state estimate and I will know your sensor characteristics, including your measurements*") has potential applications in, e.g., autonomous calibration of sensors, fault diagnosis, and detecting Bayesian behavior in agents. In model-based fault-detection [5, 6], sensor information together with solutions to related inverse filtering problems are used to detect abnormal behavior. (As trivial examples; *i)* if the true sequence of observations is known from a redundant sensor, it can be compared to the reconstructed sequence; if there is a miss-match, something is wrong, or *ii)* if multiple data batches are available, then change detection can be performed on the sequence of reconstructed observation likelihoods.) They are also of relevance in a revealed preference context in microeconomics where the aim is to detect expected utility maximization behavior of an agent; estimating the posterior given the agent's actions is a crucial step, see, e.g., [7].

Recent advances in wearables and smart-sensor technology have led to consumer grade products (smart watches with motion and heart-beat monitoring, sleep trackers, etc.) that produce vast amounts of personal data by performing state estimation. This information can serve as an indicator of health, fitness and stress. It may be very difficult, or even impossible, to access the raw sensor data since the sensor and state estimator usually are tightly integrated and encapsulated in intelligent sensor systems. Inverse filtering provides a framework for *reverse engineering* and performing *fault detection* of such sensors. In Section 5, we demonstrate our proposed solutions on a system that performs automatic sequencing of sleep stages based on *electroencephalogram* (EEG) data – the outputs of such an automatic system are exactly posteriors over the different sleep stages [8].

Another important application of the inverse filtering problem arises in *electronic warfare* and *cyber-physical security*. How can one determine how accurate an enemy's sensors are? In such problems, the state of the underlying Markov chain is usually known (a probing sequence), and one observes actions taken by the enemy which are based on filtered posterior distributions. The aim is to estimate the observation likelihood probabilities of the enemy, i.e., determine how accurate its sensors are.

**Our contributions**   It is possible to obtain a solution to the inverse filtering problem for HMMs by employing a brute-force approach (see Section 2.3) – essentially by testing observations from the alphabet, and at the same time finding system parameters consistent with the data. However, this leads to a computationally expensive combinatorial optimization problem. Instead, we demonstrate in this paper an efficient solution based on linear algebra by exploiting the inherent structure of the problem and the HMM filter. In particular, the contributions of this paper are three-fold:

1. We propose analytical solutions to three inverse filtering problems for HMMs that avoid computationally expensive *mixed integer linear program* (MILP) formulations. Moreover, we establish theorems guaranteeing unique identifiability.

2. We consider the setting where the output of the HMM filter is corrupted by noise, and propose an inverse filtering algorithm based on clustering.

3. We evaluate the algorithm on real-world data for automatic segmentation of the sleep cycle.

**Related work**   There are only two known cases where the optimal filter allows a finite dimensional characterization: the HMM filter for (discrete) HMMs, and the *Kalman filter* [9, 10] for linear Gaussian state-space models. Inverse filtering problems for the Kalman filter have been considered in, e.g., [5, 6, 10], however, inverse filtering for HMMs has, to the best knowledge of the authors, received much less attention.

The inverse filtering problem has connections to a number of other inverse problems in various fields. For example, in control theory, the fundamental *inverse optimal control problem*, whose formulation dates back to 1964 [11], studies the question: given a system and a policy, for what cost criteria is the policy optimal? In microeconomic theory, the related *problem of revealed preferences* [12] asks the question: given a set of decisions made by an agent, is it possible to determine if a utility is being maximized, and if so, which?

In machine learning, there are clear connections to, e.g., *apprenticeship learning*, *imitation learning* and *inverse reinforcement learning*, see, e.g., [13–17], which recently have received much attention. In these, the reward function of a *Markov decision process* (MDP) is learned by observing an expert demonstrating the task that an agent wants to learn to perform.

The key difference between these works and our work is the set of system parameters we aim to learn.

## 2 Preliminaries

In this section, we formulate the inverse filtering problems, discuss how these can be solved using combinatorial optimization, and state our assumptions formally. With regards to notation, all vectors are column vectors, unless transposed. The vector $\mathbb{1}$ is the vector of all ones. $^\dagger$ denotes the Moore–Penrose pseudoinverse.

### 2.1 Hidden Markov models (HMMs) and the HMM filter

We consider a discrete-time finite observation-alphabet HMM. Denote its state at time $k$ as $x_k \in \{1, \ldots, X\}$ and the corresponding observation $y_k \in \{1, \ldots, Y\}$. The underlying Markov chain $x_k$ evolves according to the row-stochastic transition probability matrix $P \in \mathbb{R}^{X \times X}$, where $[P]_{ij} = \Pr[x_{k+1} = j | x_k = i]$. The initial state $x_0$ is sampled from the probability distribution $\pi_0 \in \mathbb{R}^X$, where $[\pi_0]_i = \Pr[x_0 = i]$. The noisy observations of the underlying Markov chain are obtained from the row-stochastic observation likelihood matrix $B \in \mathbb{R}^{X \times Y}$, where $[B]_{ij} = \Pr[y_k = j | x_k = i]$ are the observation likelihood probabilities. We denote the columns of the observation likelihood matrix as $\{b_i\}_{i=1}^Y$, i.e., $B = [b_1 \ldots b_Y]$.

In the classical stochastic filtering problem, the aim is to compute the posterior distribution $\pi_k \in \mathbb{R}^X$ of the latent state (Markov chain, in our case) at time $k$, given observations from the system up to time $k$. The *HMM filter* [1, 4] computes these posteriors via the following recursive update:

$$\pi_k = \frac{B_{y_k} P^T \pi_{k-1}}{\mathbb{1}^T B_{y_k} P^T \pi_{k-1}}, \tag{1}$$

initialized by $\pi_0$, where $[\pi_k]_i = \Pr[x_k = i | y_1, \ldots, y_k]$ is the posterior distribution at time $k$, $B_{y_k} = \text{diag}(b_{y_k}) \in \mathbb{R}^{X \times X}$, and $\{y_k\}_{k=1}^N$ is a set of observations.

### 2.2 Inverse HMM filtering problem formulations

The inverse filtering problem for HMMs is not a single problem – multiple variants can be formulated depending on what information is available *a priori*. We pose and consider a number of variations of increasing levels of generality depending on what data we can extract from the sensor system. To restrict the scope of the paper, we assume throughout that the transition matrix $P$ is known, and is the same in both the system and the HMM filter (i.e, we do not consider miss-matched HMM filtering problems). Formally, the inverse filtering problems considered in this paper are as follows:

**Problem 1** (Inverse filtering problem with unknown observations). *Consider the known data $\mathcal{D} = \left\{ P, B, \{\pi_k\}_{k=0}^N \right\}$, where the posteriors have been generated by an HMM-filter sensor. Reconstruct the observations $\{y_k\}_{k=1}^N$.*

**Problem 2** (Inverse filtering problem with unknown sensor). *Consider the known data $\mathcal{D} = \left\{ P, \{y_k\}_{k=1}^N, \{\pi_k\}_{k=0}^N \right\}$, where the posteriors have been generated by an HMM-filter sensor. Reconstruct the observation likelihood matrix $B$.*

Combining these two formulations yields the general problem:

**Problem 3** (Inverse filtering problem with unknown sensor and observations). *Consider the known data $\mathcal{D} = \left\{ P, \{\pi_k\}_{k=0}^N \right\}$, where the posteriors have been generated by an HMM-filter sensor. Reconstruct the observations $\{y_k\}_{k=1}^N$ and the observation likelihood matrix $B$.*

Finally, we consider the more general setting where the posteriors we obtain are corrupted by noise (due to, e.g., quantization, measurement or model uncertainties). In particular, we consider the case where the following sequence of noisy posteriors is obtained over time:

$$\tilde{\pi}_k = \pi_k + \text{noise}, \tag{2}$$

from the sensor system. We state directly the generalization of Problem 3 (the corresponding generalizations of Problems 1 and 2 follow as special-cases):

**Problem 4** (Noise-corrupted inverse filtering problem with unknown sensor and observations). *Consider the data $\mathcal{D} = \left\{ P, \{\tilde{\pi}_k\}_{k=0}^N \right\}$, where the posteriors $\pi_k$ have been generated by an HMM-filter sensor, but we obtain noise-corrupted measurements $\tilde{\pi}_k$. Estimate the observations $\{y_k\}_{k=1}^N$ and the observation likelihood matrix $B$.*

## 2.3 Inverse filtering as an optimization problem

It is possible to formulate Problems 1-4 as optimization problems of increasing levels of generality. As a first step, rewrite the HMM filter equation (1) as:[1]

$$(1) \iff b_{y_k}^T P^T \pi_{k-1} \pi_k = \text{diag}(b_{y_k}) P^T \pi_{k-1}. \tag{3}$$

In Problem 3 we need to find what observation occurred at each time instant (a combinatorial problem), and at the same time reconstruct an observation likelihood matrix consistent with the data. To be consistent with the data, equation (3) has to be satisfied. This feasibility problem can be formulated as the following *mixed-integer linear program* (MILP):

$$
\min_{\{y_k\}_{k=1}^N, \{b_i\}_{i=1}^Y} \quad \sum_{k=1}^N \| b_{y_k}^T P^T \pi_{k-1} \pi_k - \text{diag}(b_{y_k}) P^T \pi_{k-1} \|_\infty
$$
$$
\begin{aligned}
\text{s.t.} \quad & y_k \in \{1, \dots, Y\}, && \text{for } k = 1, \dots, N, \\
& b_i \geq 0, && \text{for } i = 1, \dots, Y, \\
& [b_1 \dots b_Y] \mathbb{1} = \mathbb{1},
\end{aligned} \tag{4}
$$

where the choice of norm is arbitrary since for noise-free data it is possible to exactly fit observations and an observation likelihood matrix. In Problem 1, the $b_i$:s are dropped as optimization variables and the problem reduces to an *integer program* (IP). In Problem 2, where the sequence of observations is known, the problem reduces to a *linear program* (LP).

Despite the ease of formulation, the down-side of this approach is that, even though Problems 1 and 2 are computationally tractable, the MILP-formulation of Problem 3 can become computationally very expensive for larger data sets. In the following sections, we will outline how the problems can be solved efficiently by exploiting the structure of the HMM filter.

## 2.4 Assumptions

Before providing solutions to Problems 1-4, we state the assumptions that the HMMs in this paper need to satisfy to guarantee unique solutions. The first assumption serves as a proxy for ergodicity of the HMM and the HMM filter – it is a common assumption in statistical inference for HMMs [18, 4].

**Assumption 1** (Ergodicity). *The transition matrix $P$ and the observation matrix $B$ are elementwise (strictly) positive.*

The second assumption is a natural rank assumption on the observation likelihoods. The assumption says that the conditional distribution of any observation is not a linear combination of the conditional distributions of any other observations.

**Assumption 2** (Distinguishable observation likelihoods). *The observation likelihood matrix $B$ is full column rank.*

We will see that this assumption can be relaxed to the following assumption in problems where only the sequence of observations is to be reconstructed:

**Assumption 3** (Non-parallel observation likelihoods). *No pair of columns of the observation likelihood matrix $B$ is colinear, i.e., $b_i \neq \kappa b_j$ for any real number $\kappa$ and any $i \neq j$.*

Without Assumption 3, it is impossible to distinguish between observation $i$ and observation $j$. Note also that Assumption 2 implies Assumption 3.

## 3 Solution to the inverse filtering problem for HMMs in absence of noise

In this section, we detail our solutions to Problems 1-3. We first provide the following two useful lemmas that will be key to the solutions for Problems 1-4. They give an alternative characterization of the HMM-filter update equation. (Note that all proofs are in the supplementary material.)

**Lemma 1.** *The HMM-filter update equation (3) can equivalently be written*

$$\left(\pi_k(P^T\pi_{k-1})^T - \mathrm{diag}(P^T\pi_{k-1})\right)b_{y_k} = 0. \tag{5}$$

The second lemma characterizes the solutions to (5).

**Lemma 2.** *Under Assumption 1, the nullspace of the $X \times X$ matrix*

$$\pi_k(P^T\pi_{k-1})^T - \mathrm{diag}(P^T\pi_{k-1}) \tag{6}$$

*is of dimension one for $k > 1$.*

### 3.1 Solution to the inverse filtering problem with unknown observations

In the formulation of Problem 1, we assumed that the observation likelihoods $B$ were known, and aimed to reconstruct the sequence of observations from the posterior data. Equation (5) constrains which columns of the observation matrix $B$ that are consistent with the update of the posterior vector at each time instant. Formally, any sequence

$$\hat{y}_k \in \left\{y \in \{1,\ldots,Y\} : \left(\pi_k(P^T\pi_{k-1})^T - \mathrm{diag}(P^T\pi_{k-1})\right)b_y = 0\right\}, \tag{7}$$

for $k = 1,\ldots,N$, is consistent with the HMM filter posterior updates. (Recall that $b_y$ denotes column $y$ of the observation matrix $B$.) Since the problems (7) are decoupled in time $k$, they can trivially be solved in parallel.

**Theorem 1.** *Under Assumptions 1 and 3, the set in the right-hand side of equation (7) is a singleton, and is equal to the true observation, i.e.,*

$$\hat{y}_k = y_k, \tag{8}$$

*for $k > 1$.*

### 3.2 Solution to the inverse filtering problem with unknown sensor

The second inverse filtering problem we consider is when the sequence of observations is known, but the observation likelihoods $B$ are unknown (Problem 2). This problem can be solved by exploiting Lemmas 1 and 2.

Computing a basis for the nullspace of the coefficient matrix in formulation (5) of the HMM filter recovers, according to Lemmas 1 and 2, *the direction* of one column of $B$. In particular, the direction of the column corresponding to observation $y_k$, i.e., $b_{y_k}$. From such basis vectors, we can construct a matrix $C \in \mathbb{R}^{X \times Y}$ where the $y$th column is aligned with $b_y$. Note that to be able to fully construct this matrix, every observation from the set $\{1,\ldots,Y\}$ needs to have been observed at least once.

Due to being basis vectors for nullspaces, the columns of $C$ are only determined up to scalings, so we need to exploit the structure of the observation matrix $B$ to properly normalize them. To form an estimate $\hat{B}$ from $C$, we employ that the observation likelihood matrix is row-stochastic. This means that we should rescale each column:

$$\hat{B} = C\,\mathrm{diag}(\alpha) \tag{9}$$

for some $\alpha \in \mathbb{R}^Y$, such that $\hat{B}\mathbb{1} = \mathbb{1}$. Details are provided in the following theorem.

**Theorem 2.** *If Assumption 1 holds, and every possible observation has been observed (i.e., that $\{1,\ldots,Y\} \subset \{y_k\}_{k=1}^N$), then:*

   *i) there exists $\alpha \in \mathbb{R}^Y$ such that $\hat{B} = B$,*

   *ii) if Assumption 2 holds, then the choice of $\alpha$ is unique, and $\hat{B}$ is equal to $B$. In particular, $\alpha = C^\dagger \mathbb{1}$.*

### 3.3 Solution to the inverse filtering problem with unknown sensor and observations

Finally, we turn to the general formulation in which we consider the combination of the previous two problems: both the sequence of observations and the observation likelihoods are unknown (Problem 3). Again, the solution follows from Lemmas 1 and 2. Note that there will be a degree of freedom since we can arbitrarily relabel each observation and correspondingly permute the columns of the observation likelihood matrix.

As in the solution to Problem 2, computing a basis vector, say $\bar{c}_k$, for the nullspace of the coefficient matrix in equation (5) recovers the direction of one column of the $B$ matrix. However, since the sequence of observations is unknown, we do not know which column. To circumvent this, we concatenate such basis vectors in a matrix[2]

$$\bar{C} = [\bar{c}_2 \ldots \bar{c}_N] \in \mathbb{R}^{X \times (N-1)}. \tag{10}$$

For sufficiently large $N$ – essentially when every possible observation has been processed by the HMM filter – the matrix $\bar{C}$ in (10) will contain $Y$ columns out of which no pair is colinear (due to Assumption 3). All the columns that are parallel correspond to one particular observation. Let $\{\sigma_1, \ldots, \sigma_Y\}$ be the indices of $Y$ such columns, and construct

$$C = \bar{C}\Sigma \tag{11}$$

using the *selection matrix*

$$\Sigma = [e_{\sigma_1} \ldots e_{\sigma_Y}] \in \mathbb{R}^{(N-1) \times Y}, \tag{12}$$

where $e_i$ is the $i$th Cartesian basis vector.

**Lemma 3.** *Under Assumption 1 and Assumption 3, the expected number of samples needed to be able to construct the selection matrix $\Sigma$ is upper-bounded by*

$$\beta^{-1} \left(1 + 1/2 + \cdots + 1/Y\right), \tag{13}$$

*where $B \geq \beta > 0$ elementwise.*

With $C$ constructed in (11), we have obtained the direction of each column of the observation matrix. However, as before, they need to be properly normalized. For this, we exploit the sum-to-one property of the observation matrix as in the previous section. Let

$$\hat{B} = C \operatorname{diag}(\alpha), \tag{14}$$

for $\alpha \in \mathbb{R}^Y$, such that $\hat{B}\mathbb{1} = \mathbb{1}$. Details on how to find $\alpha$ are provided in the theorem below.

This solves the first part of the problem, i.e., reconstructing the observation matrix. Secondly, to recover the sequence of observations, take

$$\hat{y}_k \in \left\{y \in \{1, \ldots, Y\} : \hat{b}_y = \kappa \bar{c}_k \text{ for some real number } \kappa\right\}, \tag{15}$$

for $k > 1$. In words; check which columns of $\hat{B}$ that the nullspace of the HMM filter coefficient-matrix (6) is colinear with at each time instant.

**Theorem 3.** *If Assumptions 1 and 3 hold, and the number of samples $N$ is sufficiently large – see Lemma 3 – then:*

    *i) there exists $\alpha \in \mathbb{R}^Y$ in equation (14) such that $\hat{B} = B\mathcal{P}$, where $\mathcal{P}$ is a permutation matrix.*

    *ii) the set on the right-hand side of equation (15) is a singleton. Moreover, the reconstructed observations $\hat{y}_k$ are, up to relabellings corresponding to $\mathcal{P}$, equal to the true observations $y_k$.*

    *iii) if Assumption 2 holds, then the choice of $\alpha$ is unique, and $\hat{B} = B\mathcal{P}$. In particular, $\alpha = C^\dagger \mathbb{1}$.*

# 4 Solution to the inverse filtering problem for HMMs in presence of noise

In this section, we discuss the more general setting where the posteriors obtained from the sensor system are corrupted by noise. We will see that this problem naturally fits in a clustering framework since every posterior update will provide us with a noisy estimate of the direction of one column of the observation likelihood matrix. We consider an additive noise model of the following form:

**Assumption 4** (Noise model). *The posteriors are corrupted by additive noise $w_k$:*

$$\tilde{\pi}_k = \pi_k + w_k, \tag{16}$$

*such that $\mathbb{1}^T \tilde{\pi}_k = 1$ and $\tilde{\pi}_k > 0$.*

This noise model is valid, for example, when each observed posterior vector has been subsequently renormalized after noise that originates from quantization or measurement errors has been added.

In the solution proposed in Section 3.3 for the noise-free case, the matrix $\bar{C}$ in equation (10) was constructed by concatenating basis vectors for the nullspaces of the coefficient matrix in equation (5). With perturbed posterior vectors, the corresponding system of equations becomes

$$\left( \tilde{\pi}_k (P^T \tilde{\pi}_{k-1})^T - \text{diag}(P^T \tilde{\pi}_{k-1}) \right) \tilde{c}_k = 0, \tag{17}$$

where $\tilde{c}_k$ is now a perturbed (and scaled) version of $b_{y_k}$. That this equation is valid is guaranteed by the generalization of Lemma 2:

**Lemma 4.** *Under Assumptions 1 and 4, the nullspace of the matrix*

$$\tilde{\pi}_k (P^T \tilde{\pi}_{k-1})^T - \text{diag}(P^T \tilde{\pi}_{k-1}) \tag{18}$$

*is of dimension one for $k > 1$.*

**Remark 1.** *In case Assumption 4 does not hold, the problem can instead be interpreted as a perturbed eigenvector problem. The vector $\tilde{c}_k$ should then be taken as the eigenvector corresponding to the smallest eigenvalue.*

Lemma 4 says that we can construct a matrix $\tilde{C}$ (analogous to $\bar{C}$ in Section 3.3) by concatenating the basis vectors from the one-dimensional nullspaces in (17). Due to the perturbations, every solution to equation (17) will be a perturbed version of the solution to the corresponding noise-free version of the equation. This means that it will not be possible to construct a selection matrix $\Sigma$ as was done for $\bar{C}$ in equation (12). However, because there are only $Y$ unique solutions to the noise-free equations (5), it is natural to circumvent this (assuming that the perturbations are small) by clustering the columns of $\tilde{C}$ into $Y$ clusters. As the columns of $\tilde{C}$ are only unique up to scaling, the clustering has to be performed with respect to their angular separations (using, e.g., the *spherical k-means algorithm* [19]).

Let $C \in \mathbb{R}^{X \times Y}$ be the matrix of the $Y$ centroids resulting from running a clustering algorithm on the columns of $\tilde{C}$. Each centroid can be interpreted as a noisy estimate of one column of the observation likelihood matrix. To obtain a properly normalized estimate of the observation likelihood matrix, we take

$$\hat{B} = CA, \tag{19}$$

where $A \in \mathbb{R}^{Y \times Y}$. Note that, since $C$ now contains noisy estimates of the directions of the columns of the observation likelihood matrix, we are not certain to be able to properly normalize it by purely rescaling each column (i.e., taking $A$ to be a diagonal matrix as was done in Sections 3.2 and 3.3). A logical choice is the solution to the following LP,

$$\begin{aligned} \min_{A \in \mathbb{R}^{Y \times Y}} \quad & \max_{i \neq j} \left| [A]_{ij} \right| \\ \text{s.t.} \quad & CA \geq 0, \\ & CA\mathbb{1} = \mathbb{1}, \end{aligned} \tag{20}$$

which tries to minimize the off-diagonal elements of $A$. The resulting rescaling matrix $A$ guarantees that $\hat{B} = CA$ is a proper stochastic matrix (non-negative and has row-sum equal to one), as well as that the discrepancy between the directions of the columns of $C$ and $\hat{B}$ are minimized.

The second part of the problem – reconstructing the sequence of observations – follows naturally from the clustering algorithm; an estimate of the sequence is obtained by checking to what cluster the solution $\tilde{c}_k$ of equation (17) belongs in for each time instant.

# 5 Experimental results for sleep segmentation

In this section, we illustrate the inverse filtering problem on real-world data.

**Background**    Roughly one third of a person's life is spent sleeping. Sleep disorders are becoming more prevalent and, as public awareness has increased, the usage of sleep trackers is becoming wide-spread. The example below illustrates how the inverse filtering formulation and associated algorithms can be used as a step in real-time diagnosis of failure of sleep-tracking medical equipment.

During the course of sleep, a human transitions through five different sleep stages [20]: *wake*, *S1*, *S2*, *slow wave sleep* (SWS) and *rapid eye movement* (REM). An important part of sleep analysis is obtaining a patient's evolution over these sleep stages. Manual sequencing from all-night *polysomnographic* (PSG) recordings (including, e.g., *electroencephalogram* (EEG) readings) can be performed according to the *Rechtschaffen and Kales* (R&K) rules by well-trained experts [8, 20]. However, this is costly and laborious, so several works, e.g., [8, 20, 21], propose automatic sequencing based on HMMs. These systems usually output a posterior distribution over the sleep stages, or provide a Viterbi path.

A malfunction of such an automatic system could have problematic consequences since medical decisions would be based on faulty information. The inverse filtering problem arises naturally for such reasons of fault-detection. Joint knowledge of the transition matrix can be assumed, since it is possible to obtain, from public sources, manually labeled data from which an estimate of $P$ can be computed.

**Setup**    A version of the automatic sleep-staging system in [8, 20] was implemented. The mean frequency over the 0-30 Hz band of the EEG (over C3-A2 or C4-A1, according to the international 10-20 system) was used as observations. These readings were encoded to five symbols using a vector-quantization based codebook. The model was trained on data from nine patients in the PhysioNet CAP Sleep Database [22, 23]. The model was then evaluated on another patient – see Fig. 1 – over one full-night of sleep. The manually labeled stages according to K&R-rules are dashed-marked in the figure. To summarize the resulting posterior distributions over the sleep stages, we plot the mean state estimate when equidistant numbers have been assigned to each state.

For the inverse filtering, the full posterior vectors were elementwise corrupted by Gaussian noise of standard deviation $\sigma$, and projected back to the simplex (to ensure a valid posterior probability vector) – simulating a noisy reading from the automatic system. A total of one hundred noise realizations were simulated. The noise can be a manifestation of measurement or quantization noise in the sensor system, or noise related to model uncertainties (in this case, an error in the transition probability matrix $P$).

**Results**    After permuting the labels of the observations, the error in the reconstructed observation likelihood matrix, as well as the fraction of correctly reconstructed observations, were computed. This is illustrated in Fig. 2. For the 1030 quantized EEG samples from the patient, the entire procedure takes less than one second on a 2.0 Ghz Intel Core 2 Duo processor system.

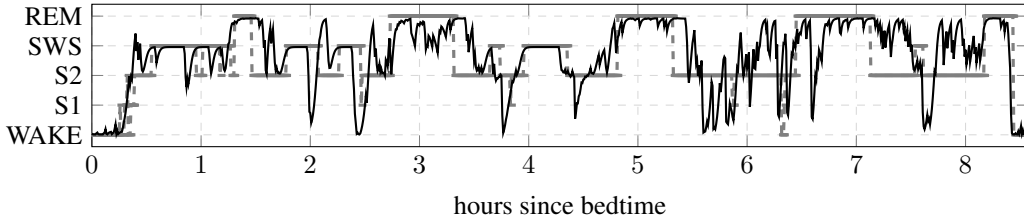

hours since bedtime

Figure 1: One night of sleep in which *polysomnographic* (PSG) observation data has been manually processed by an expert sleep analyst according to the R&K rules to obtain the sleep stages (- - - -). The posterior distribution over the sleep stages, resulting from an automatic sleep-staging system, has been summarized to a mean state estimate (———).

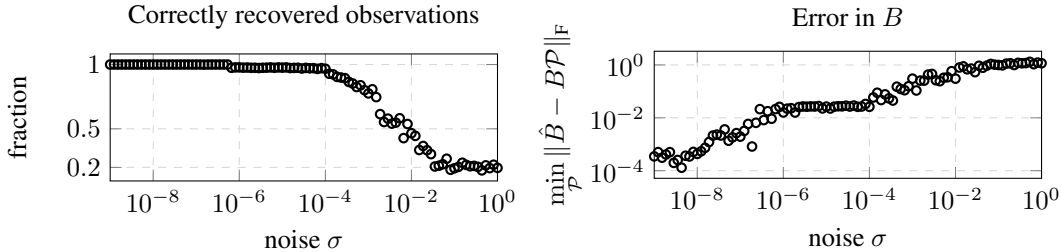

Figure 2: Result of inverse filtering for various noise standard deviations $\sigma$. The vector of posterior probabilities is perturbed elementwise with Gaussian noise. *Right:* Error in the recovered observation likelihood matrix after permuting the columns to find the best match to the true matrix. *Left:* Fraction of correctly reconstructed observations. As the signal-to-noise ratio increases, the inverse filtering algorithm successfully reconstructs the sequence of observations and estimates the observation likelihoods.

From Fig. 2, we can see that as the variance of the noise decreases, the left hand side of equation (17) converges to that of equation (5) and the true quantities are recovered. On the other extreme, as the signal-to-noise ratio becomes small, the estimated sequence of observations tends to that of a uniform distribution at $1/Y = 0.2$. This is because the clusters in $\tilde{C}$ become heavily intertwined. The discontinuous nature of the solution of the clustering algorithm is apparent by the plateau-like behaviour in the middle of the scale – a few observations linger on the edge of being assigned to the correct clusters.

In conclusion, the results show that it is possible to estimate the observation sequence processed by the automatic sleep-staging system, as well as, its sensor's specifications. This is an important step in performing fault detection for such a device: for example, using several nights of data, it is possible to perform change detection on the observation likelihoods to detect if the sleep monitoring device has failed.

## 6    Conclusions

In this paper, we have considered several inverse filtering problems for HMMs. Given posteriors from an HMM filter (or more generally, noisily observed posteriors), the aim was to reconstruct the observation likelihoods and also the sample path of observations. It was shown that a computationally expensive solution based on combinatorial optimization can be avoided by exploiting the algebraic structure of the HMM filter. We provided solutions to the inverse filtering problems, as well as theorems guaranteeing unique identifiability. The more general case of noise-corrupted posteriors was also considered. A solution based on clustering was proposed and evaluated on real-world data based on a system for automatic sleep-staging from EEG readings.

In the future, it would be interesting to consider other variations and generalizations of inverse filtering. For example, the case where the system dynamics are unknown and need to be estimated, or when only actions based on the filtered distribution can be observed.

**Acknowledgments**

This work was partially supported by the Swedish Research Council under contract 2016-06079, the U.S. Army Research Office under grant 12346080 and the National Science Foundation under grant 1714180. The authors would like to thank Alexandre Proutiere for helpful comments during the preparation of this work.

## Footnotes

[1]Multiplication by the denominator is allowed under Assumption 1 – see below.

[2]We start with $\bar{c}_2$, since we make no assumption on the positivity of $\pi_0$ – see the proof of Lemma 2.

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
