[Supplementary Material]

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

# A   Proofs

In this appendix, we provide proofs that were omitted in the paper.

## A.1   Proof of Lemma 1

*Proof.* Consider row $i$ of equation (3):

$$[b_{y_k}^T P^T \pi_{k-1} \pi_k]_i = [\text{diag}(b_{y_k}) P^T \pi_{k-1}]_i \qquad \Longleftrightarrow$$

$$\sum_{j=1}^{X} [b_{y_k}]_j [P^T \pi_{k-1}]_j [\pi_k]_i = \sum_{j=1}^{X} [\text{diag}(b_{y_k})]_{ij} [P^T \pi_{k-1}]_j \qquad \Longleftrightarrow$$

$$\sum_{j=1}^{X} [b_{y_k}]_j [P^T \pi_{k-1}]_j [\pi_k]_i = \sum_{j=1}^{X} \delta_{ij} [b_{y_k}]_j [P^T \pi_{k-1}]_j \qquad \Longleftrightarrow$$

$$\sum_{j=1}^{X} \left( [P^T \pi_{k-1}]_j [\pi_k]_i - \delta_{ij} [P^T \pi_{k-1}]_j \right) [b_{y_k}]_j = 0 \qquad \Longleftrightarrow$$

$$\left[ \left( \pi_k (P^T \pi_{k-1})^T - \text{diag}(P^T \pi_{k-1}) \right) b_{y_k} \right]_i = 0, \qquad (21)$$

where $\delta_{ij}$ is equal to one if $i = j$, and zero otherwise. $\qquad \square$

## A.2   Proof of Lemma 2

*Proof.* For $k > 1$, $\pi_{k-1} > 0$ under the assumptions of positive $P$ and $B$. Hence, $\text{diag}(P^T \pi_{k-1})$ is a non-singular matrix. The term $\pi_k (P^T \pi_{k-1})^T = \pi_k \pi_{k-1}^T P$ is a rank-1 update. Therefore,[3]

$$\text{rank}(\pi_k (P^T \pi_{k-1})^T - \text{diag}(P^T \pi_{k-1})) \geq X - 1. \qquad (22)$$

However, since

$$\mathbb{1}^T \left( \pi_k (P^T \pi_{k-1})^T - \text{diag}(P^T \pi_{k-1}) \right) = (P^T \pi_{k-1})^T - (P^T \pi_{k-1})^T = 0, \qquad (23)$$

we have that

$$\text{rank}(\pi_k (P^T \pi_{k-1})^T - \text{diag}(P^T \pi_{k-1})) \leq X - 1. \qquad (24)$$

$\square$

## A.3   Proof of Theorem 1

*Proof.* The true observation is, of course, consistent with the observed data, so it will be an element of the set in (7). From Lemma 2, we know that the only solutions (with respect to $b_y$) consistent with the data lie on a one-dimensional subspace. However, since no pair of columns of $B$ are colinear (by Assumption 3), a unique column of $B$ will fulfill the equation – implying that the set is singleton.

$\square$

$$\text{rank}(A - B) \geq \text{rank}(A) - \text{rank}(B),$$

where, in this case, $\text{rank}(A) = X$ (full rank) and $\text{rank}(B) = 1$ (rank-1 update). This inequality can be derived by replacing $A$ by $(A - B)$ in the well-known inequality

$$\text{rank}(A + B) \leq \text{rank}(A) + \text{rank}(B),$$

as follows:

$$\text{rank}((A - B) + B) \leq \text{rank}(A - B) + \text{rank}(B) \implies$$
$$\text{rank}(A) \leq \text{rank}(A - B) + \text{rank}(B) \implies$$
$$\text{rank}(A - B) \geq \text{rank}(A) - \text{rank}(B).$$

*Remark:* Since we make no assumption on the positivity of $\pi_0$, we can not formally guarantee that we can recover the first observation $y_1$ uniquely; the dimension of the nullspace of the matrix in Lemma 2 can be larger than one.

## A.4  Proof of Theorem 2

*Proof.* The matrix $C$ is constructed in such a way that, by Lemmas 1 and 2, every column of $C$ is a scaled version of the corresponding column of $B$. This implies that there exists a set of unique numbers $\alpha_i^* \neq 0$, such that $b_i = \alpha_i^* c_i$ for $i = 1, \ldots, Y$, where $c_i$ denotes column $i$ of $C$. In vector notation, this means that there exists a unique $\alpha^* \in \mathbb{R}^Y$ such that

$$C \operatorname{diag}(\alpha^*) = B. \tag{25}$$

Multiplying this equation from the right by $\mathbb{1}$, we obtain

$$
\begin{aligned}
C \operatorname{diag}(\alpha^*) &= B && \Longrightarrow \\
C \operatorname{diag}(\alpha^*)\mathbb{1} = B\mathbb{1} &= \mathbb{1} && \Longleftrightarrow \\
C\alpha^* &= \mathbb{1}. &&
\end{aligned}
\tag{26}
$$

*Proof of i):* To normalize our estimate $\hat{B} = C \operatorname{diag}(\alpha)$, we seek an $\alpha$ that fulfills the condition $\hat{B}\mathbb{1} = \mathbb{1}$. This is equivalent to finding an $\alpha$ fulfilling

$$C \operatorname{diag}(\alpha)\mathbb{1} = C\alpha = \mathbb{1}. \tag{27}$$

Since $\alpha^*$ solves this equation, the existence of a solution is guaranteed.

*Proof of ii):* The solution to

$$C\alpha = \mathbb{1} \tag{28}$$

is unique if and only if $C$ has full column rank. From equation (25) and the fact that $\alpha^*$ has non-zero elements, this is equivalent to $B$ having full column rank. However, $B$ has full column rank by Assumption 2. The unique solution can in this case be obtained as

$$\alpha = (C^T C)^{-1} C^T \mathbb{1} = C^\dagger \mathbb{1}. \tag{29}$$

Since the solution to equation (28) is unique, and $\alpha^*$ is a solution by equation (26), we conclude that $\alpha = \alpha^*$, so that $\hat{B} = C \operatorname{diag}(\alpha) = C \operatorname{diag}(\alpha^*) = B$. $\qquad\square$

## A.5  Proof of Lemma 3

*Proof.* To be able to construct the selection matrix $\Sigma$, every observation from the set $\{1, \ldots, Y\}$ needs to have been observed at least once. Since $B \geq \beta$ elementwise, each observation will be sampled at every time instant with at least probability $\beta$, independently of what state the system is in.

We upper bound the expected time it takes to have observed all observations with the following i.i.d process (which can be interpreted as a variation of the *coupon collector's* problem, e.g., [24]). At every time instant we, either, *i)* obtain observation $i$ with probability $\beta$ (for $i = 1, \ldots, Y$), or, *ii)* obtain no observation at all, with probability $1 - Y\beta$.

Let $N$ denote the number of samples it takes in this process to have seen all the $Y$ unique observations. Let $n_i$ denote the number of samples it takes until a new unique observation is seen, after the $(i-1)$th was observed. After having observed $i - 1$ unique observations, the probability of sampling a new unique observation is

$$p_i = Y\beta \times \frac{1}{Y} (Y - (i - 1)) = (Y - (i - 1))\, \beta. \tag{30}$$

Every $n_i$ follows a geometric distribution with success probability $p_i$. Hence,

$$
\begin{aligned}
\mathbb{E}\{N\} &= \mathbb{E}\{n_1 + \cdots + n_Y\} \\
&= \sum_{i=1}^{Y} \frac{1}{p_i} \\
&= \frac{1}{\beta} \sum_{i=1}^{Y} \frac{1}{Y - (i - 1)} \\
&= \beta^{-1}\left(1 + \cdots + 1/Y\right).
\end{aligned}
\tag{31}
$$

This upper bounds the number of samples, since the probability of sampling each observation is in fact greater than (or equal to) $\beta$.

$\square$

## A.6 Proof of Theorem 3

*Proof.* By Lemmas 1 and 2, every column of $\bar{C}$ will be a scaled version of one column of the observation matrix $B$. The selection matrix $\Sigma$ picks $Y$ of these columns, where no pair is colinear.[4] Since no two columns of $B$ are parallel – by Assumption 3 – this means that $C = \bar{C}\Sigma$ will contain all $Y$ columns of $B$, but scaled and permuted. Formally,

$$B\mathcal{P} = C \operatorname{diag}(\alpha^*), \tag{32}$$

for some permutation matrix $\mathcal{P}$, which is decided by the choice of $\Sigma$, and a (for this $\mathcal{P}$) unique $\alpha^* \in \mathbb{R}^Y$ with non-zero elements.

Multiplying this equation from the right by $\mathbb{1}$ yields

$$
\begin{aligned}
B\mathcal{P} &= C \operatorname{diag}(\alpha^*) && \Longrightarrow \\
B\mathcal{P}\mathbb{1} &= C \operatorname{diag}(\alpha^*)\mathbb{1} && \Longleftrightarrow \\
B\mathbb{1} &= C\alpha^* && \Longleftrightarrow \\
\mathbb{1} &= C\alpha^*.
\end{aligned}
\tag{33}
$$

*Proof of i)* To normalize our estimate $\hat{B}$, we seek an $\alpha$ that fulfills the condition

$$\mathbb{1} = \hat{B}\mathbb{1} = C \operatorname{diag}(\alpha)\mathbb{1} = C\alpha, \tag{34}$$

From (33), we have that $\alpha^*$ is a solution. This guarantees existence.

*Proof of ii)* The $\hat{B}$ matrix is constructed from $Y$ columns where no pair is colinear. Hence, the nullbasis $\bar{c}_k$ can at most be parallel to one of its columns. This implies that the set is a singleton.

Lemmas 1 and 2, together with the fact that no two columns of $B$ are colinear, imply that a single column of $B$ is in the one-dimensional subspace of the matrix in equation (5). Since the columns of $\hat{B}$ are scaled and permuted (according to $\mathcal{P}$) versions of those in $B$, the true sequence of observations is obtained by relabeling the estimated sequence according to $\mathcal{P}$.

*Proof of iii)* The equation we seek to solve to normalize $\hat{B}$,

$$C\alpha = \mathbb{1}, \tag{35}$$

has a unique solution if and only if $C$ has full column rank. $C$ has full column rank since it is constructed, essentially, by permuting and scaling the columns of $B$ (which has full column rank by Assumption 2). The unique solution is given by

$$\alpha = (C^T C)^{-1} C^T \mathbb{1} = C^\dagger \mathbb{1}. \tag{36}$$

Moreover, since $\alpha^*$ is a solution – by equation (33) – we conclude that the unique $\alpha = \alpha^*$, and hence that

$$\hat{B} = C \operatorname{diag}(\alpha) = C \operatorname{diag}(\alpha^*) = B\mathcal{P}. \tag{37}$$

$\square$

## A.7 Proof of Lemma 4

*Proof.* The proof is identical to that of Lemma 2. $\square$

Figure 3: The observation data corresponding to Fig. 1. *Top:* Continuously valued observations of the EEG spectral frequency. *Bottom:* Corresponding discrete observation-data from a quantization codebook.

# B  Description of the automatic sleep-staging system

As described in detail in [20], the time-series EEG data was divided into segments of length 30 seconds. The power spectra of the 15 non-overlapping 2-second sub-windows were then computed and averaged to obtain a smoothed power spectrum $PS(\cdot)$. The *spectral frequency*, defined as $SF = \sum_{j=0}^{30} j PS(j) / \sum_{j=0}^{30} PS(j)$, was taken as the observation for each 30 second interval. This was computed using data from nine different patients from the PhysioNet CAP Sleep Database [22, 23]. The resulting time-series were subsequently concatenated, and a *k-means* algorithm was applied to obtain a codebook of size five.

The same procedure was then performed on another patient, yielding the $N = 1030$ continuously valued observations in the top plot of Fig. 3. Every sample is spaced 30 seconds apart. The same codebook was used to quantize the data – the result can be seen in the lower plot. This is the observation sequence on which the HMM filter was run to obtain Fig. 1.

The transition matrix $P$ was computed as the maximum-likelihood estimate from manually annotated state data (also from the PhysioNet CAP Sleep Database [22, 23]). The observation matrix $B$ was computed using the *expectation-maximization* (EM) algorithm on the quantized observation-data constructed above for the nine patients.