[Reviews · NeurIPS 2017]

Reviewer 1



The paper addresses recovery of the observation sequence given known posterior state estimates, but unknown observations and/or sensor model and also in an extension, noise-corrupted measurements. There is a nice progression of the problem through IP, LP, and MILP followed by a more careful analytical derivation of the answers in the noise-free case, and a seemingly approximate though empirically effective approach (cf. Fig 2) using spherical K-means as a subroutine in the noisy case. Honestly, most of the motivations seem to be unrealistic, especially the cyber-physical security setting where one does not observe posteriors, but simply an action based on a presumed argmax w.r.t. posteriors. The EEG application (while somewhat narrow) seems to be the best motivation, however, the sole example is to compare resconstructed observations to a redundant method of sensing -- is this really a compelling application? Is it actually used in practice? Some additional details are provided on page 8, but this is not enough for me to fully understand the rationale for the approach. One is given posterior state estimates of sleep stage for a new patient and the goal is to provide the corresponding mean EEG frequency observations? I don't see the point. Minor terminological question: I've always viewed the HMM to be a type of sequential graphical model structure allowing any type of random variable (discrete or continuous) of which discrete HMMs and Kalman filters (continuous Gaussian) are special cases. Does the terminology "HMM" really assume discrete random variables? Overall, though the motivation for the work is not very strong for me, the results in this paper address a novel problem, and make technically innovative and honestly surprising insights in terms of the tractability and uniqueness of the recovery problems addressed in the paper. I see this paper as a very technically interesting solution that may find future application in use cases the authors have not considered.

Reviewer 2



The paper describes the solutions to three separate (related) inverse problems for the HMM, and then applies this to polysonmography data for automatic sleep staging. In general, the paper is well written. The method appears to be novel, although I don't have extensive knowledge of this area. The weakest part of the paper is the experimental section: a single set of experiments is performed (with testing on a separate subject), whilst varying the noise. This does show that the method works for this particular application, but doesn't show how generally applicable the method is. Also, I'm not sure why the posteriors would be corrupted by noise? What's the use case? Specific comments - L16 transitions according to the row-stochastic: missing "are sampled"? - last line in eq 4 - column vector times column vector equals column vector?? B1 = 1?

Reviewer 3



This is a well written paper considering an inverse filtering problem in HMMs. The material is easy to follow and makes sense, as well as the experimental section. As far as I know this problem has not been previously addressed. My main criticism is that the experimental section is there mainly to show that the algorithms performs as promised, as opposed to backing up the original motivation for considering the inverse filtering problem in the first place. I see that the inverse filtering problem in HMMs might be useful, but the experimental results provided do not reinforce it in any way. For example, are there any examples that the HMM sensing mechanism fails? Could those example be simulated in some reasonable way and then the proposed algorithm actually used to perform the fault detection? Anyhow, I still consider this paper as a valuable material for NIPS. Other comments: Eq. (22) in the supp. material - I’m not sure I’m following the reasoning here. Is there some kind of inequality that is used to show that the sum of two matrices is no less than …? I understand that there are no statistical assumptions on the noise? (e.g. iid?) Could some of those assumptions lead to theoretical results from the clustering algorithm perhaps?